# Modelling geographical and built environment's attributes as predictors of human vulnerability during tsunami evacuations: a multi-case study and paths to improvement

Jorge León[1], Alejandra Gubler[2], Alonso Ogueda[3]

[1]Departamento de Arquitectura, Universidad Técnica Federico Santa María, Valparaíso, Chile
[2]Research Center for Integrated Disaster Risk Management (CIGIDEN), Santiago, Chile
[3]Department of Mathematical Sciences, George Mason University, Virginia, United States

*Correspondence to*: Jorge León (jorge.leon@usm.cl)

**Abstract.** Evacuation is the most important and effective method to save human lives during a tsunami. In this respect, challenges exist in developing quantitative analyses of the relationships between the evacuation potential and the built environment and geographical attributes of coastal locations. This paper proposes a computer-based modelling approach (including inundation, evacuation, and built environment metrics), followed by multivariate regressive analysis, to estimate how those attributes might influence the expected tsunami death ratios of seven Chilean coastal cities. We obtained, for the examined variables, their average values to different thresholds of the death ratio. Also, our statistical analysis allowed us to compare the relative importance of each metric, showing that the maximum flood, the straightness of the street network, the total route length, and the travel time can have a significant impact on the expected death ratios. Moreover, we suggest that these results could lead to spatial planning guidelines for developing new urban areas into exposed territories (if this expansion cannot be restricted or discouraged) or retrofitting existing ones, with the final aim of enhancing evacuation and therefore increasing resilience.

## 1 Introduction

Tsunamis are relatively rare phenomena but capable of triggering widespread destruction and causing significant human casualties in exposed coastal areas. In the last two decades, devastating events, including those in Indonesia (2004, 2006, 2010, 2018), Samoa (2009), Chile (2010, 2014, 2015) and Japan (2011), provoked more than 250,000 deaths globally (WHO, 2021). Authorities and scholars have suggested and developed a range of integrated countermeasures to reduce tsunami risk: 'hard' strategies like structural defences (e.g. sea walls, breakwaters, flood gates, and control forests) and the construction of elevated ground, and 'soft' approaches focused on education and policy, like land-use and built-environment planning, plus early warning and emergency management systems (Koshimura and Shuto, 2015; Suppasri et al., 2012b, 2013; Ting et al., 2015; Tsimopoulou et al., 2012). While 'hard' countermeasures are uncommon out of Japan, 'soft' planning-focused strategies require extended periods and high political and community support to be implemented.

Typically, this support (plus its necessary technical and monetary resources) is hard to achieve in developing countries like Indonesia and Chile (with other urgent everyday needs), where hence community-focused emergency management, emphasising evacuation, is the most feasible strategy to reduce the vulnerability of populations to tsunamis. Moreover, there is a growing consensus on evacuation as the most important and effective method for saving human lives during a tsunami (Shuto, 2005; Suppasri et al., 2012b), which is particularly true in areas exposed to near-field events, with peak arrival times as short as 15 min, as it was shown for the Chilean 2014 and 2015 events by Catalán et al. (2015) and Aránguiz et al. (2016), respectively.

In the context of disaster risk reduction policies and studies, we can define 'risk' as the potential for adverse consequences for human or ecological systems, which results from dynamic interactions between natural or manmade hazards with the exposure and vulnerability of the affected human or ecological systems (IPCC, 2020). According to the UNDRR Terminology (UNDRR, 2022), a hazard is "a process, phenomenon or human activity that may cause loss of life, injury or other health impacts, property damage, social and economic disruption or environmental degradation", while vulnerability can be defined as "the conditions determined by physical, social, economic and environmental factors or processes which increase the susceptibility of an individual, a community, assets or systems to the impacts of hazards", and exposure identifies "the situation of people, infrastructure, housing, production capacities and other tangible human assets located in hazard-prone areas". Authors like Birkmann (2006) and Frazier et al. (2014) stress the need for strengthening and focus risk mitigation and adaptation plans through the spatial assessment of hazard, vulnerability, and exposure factors. In line with this, based on thorough analyses of previous tsunamis disasters' outcomes (including the 2011 Great East Japan Earthquake and Tsunami, the 2010 Chilean tsunami, the 2009 Samoan tsunami, and the 2004 Indian Ocean tsunami) or pre-disaster modelling, scholars like Anwar et al. (2011), Birkmann et al. (2010), Eckert et al. (2012), González-Riancho et al. (2015), Suppasri et al. (2016), and Zamora et al. (2021) have underlined a range of characteristics leading to tsunami risk (with a focus on either the population or the built environment). These aspects comprise determinants of hazard (e.g. tsunami height, flow depth and arrival time), exposure (e.g. geomorphological characteristics of the inhabited areas and manmade features, including elevation, shoreline distance, number of people exposed, population density, housing density, locations of infrastructures, and types of land use) and vulnerability (e.g. warning systems, governance and institutional arrangements, evacuation potential, economic resources, education, personal awareness/knowledge/decision-making capacity).

Several studies aim at quantitatively examining tsunami vulnerability and its correlation with geographical, built environment and socio-psychological features, within a spatially specific area or domain of study, from neighbourhoods to whole regions, including blocks, districts, cities, and metropolitan areas. For instance, as shown by Tarbotton et al. (2015), most researchers use post-tsunami destruction data to focus on built structures and develop statistically-based empirical vulnerability functions that model the damage response to tsunamis. A common type of function is the fragility function, which combines the probability of damage (Y-axis) with hydrodynamic characteristics such as flood depth, flow velocity and force (X-axis). Typically, researchers develop these curves by integrating satellite remote sensing, numerical modelling of tsunami inundation, and post-tsunami survey data (examined in GIS systems) (Koshimura et al., 2009). A large group of

these studies focus on the losses in the built environment. For instance, Suppasri et al. (2012a) (using data from the 2011 Great East Japan Earthquake and Tsunami) applied least squares regression to demonstrate how building characteristics like the structural material, number of stories and coastal topography can influence their damage levels. Other tsunami disasters examined through this approach include the 2010 Chilean Tsunami in Dichato (Mas et al., 2012b) and the 2018 Sulawesi tsunami at Palu Bay in Indonesia (Mas et al., 2020). In the former, the researchers estimated the affected houses' structural

fragility through a post-tsunami survey. They combined this with an interpolated inundation depth (developed in geographic information systems from measures taken in the field) to deliver a tsunami fragility curve. In the case of the Sulawesi tsunami, the authors created the fragility functions by integrating field survey data, visual interpretation of satellite images, and machine learning for multi-sensor and multitemporal satellite images.

Other studies focus on quantitatively assessing human vulnerability to tsunamis and its possible explanatory factors. For

instance, working with the case of the 2004 tsunami disaster in Banda Aceh, Indonesia, Koshimura et al. (2009) used regressive analysis to develop a fragility function for human death ratio through the combination of tsunami modelling and post-tsunami data. This function used the number of dead, missing, and saved residents in 88 examined villages, plus the modelled inundation depth. Yun and Hamada (2015) interviewed 1,153 witnesses (and also used data for behaviour of the dead and missing) of the 2011 Great East Japan Earthquake and Tsunami to develop a conditional logistic regression model

and identify the factors that influenced life safety during that catastrophe. They found out that the fatality rate is significantly influenced by the tsunami height, aged population, speed, and region of analysis. Suppasri et al. (2016) used spatially-accurate data from areas less than 3km2 wide (with an inundation ratio greater than 70%) and statistical analysis to examine the fatality ratios and the factors that affected human fatalities during that same event. Their findings show that (in different manners depending on the region of analysis), fatality ratios are affected by the tsunami characteristics (inundation depth,

wave force, arrival time), topographical characteristics (slope, elevation, type of coast), regional characteristics (existence or absence of defense structures, warning systems and evacuation facilities), and human characteristics (existence or absence of knowledge, awareness, and decision-making capacity). Nateghi et al. (2016) analysed municipality-level and sub-municipality-level data from the 1896, 1933, 1960, and 2011 tsunamis that affected the Tohoku area in Japan. With this information, they worked out a model based on statistical learning methods that allowed them to appraise the effectiveness

of seawalls and coastal forests in mitigating destruction and death rates provoked by tsunamis. Goto and Nakasu (2018) used data from the 2011 Great East Japan Earthquake and Tsunami to propose a Human Vulnerability Index (HVI) that combines each location's fatality rate and the rate of incidence of washed-out buildings. Moreover, they applied multivariate regressive analysis to identify four explanatory variables for this index: (1) Allowance period (the tsunami arrival time divided by the distance to a safe place; (2) Preparedness (the rate of affected evacuees for analysis who had prepared emergency carry-out

bags beforehand); (3) Road serviceability (the rate of car-using evacuees × car speed); and (4) Warning effect (multiplication of announced tsunami height and cognition rate of warning). Also working in the context of the 2011 disaster, Latcharote et al. (2018) integrated surveyed fatality ratios with tsunami arrival times (obtained from flood modelling) into linear and nonlinear regression analyses to find out the relationships between them. Moreover, they examined the different findings for

two topographically different coastal areas (the Sanriku ria-coast and the Sendai plain). Their findings show that the fatality ratios decrease as the tsunami arrival times increase (for all the examined cases), and that (in the case of the Sendai plain) the fatality rates of females and those above 65 were higher than those of males and those of all ages, respectively. Lastly, Yavuz et al. (2020) used probabilistic tsunami modelling (developed from earthquake databases from 1900–2013) to evaluate social, economic and environmental risks on the Eastern Mediterranean coast. Specifically, they defined social risk as to the number of people in areas where inundation depth reaches 0.5 m or higher.

In cases where tsunamis have not occurred recently or their data is not available, researchers typically use computer-based models to estimate human vulnerability according to simulated scenarios. For instance, Sugimoto et al. (2003) developed a tsunami human damage prediction method for Usa town, Tosa City, Shikoku Island, Japan, in the context of a possible Nankai earthquake to occur during the first half of the 21st century. Their method comprised a GIS-based spatial model integrating tsunami numerical modelling, exposed populations, and expected evacuation behaviours (e.g. departure times) obtained from questionnaire surveys. This model delivered the predicted loss of human lives in 3 different scenarios, depending on the tsunami hazard factors (over 0.5 m inundation depth or more than 2.0 m/s flow velocity) and evacuation behaviour (with or without evacuation activities). In line with this, evacuation modelling has been extensively used in recent years to estimate human casualties during tsunami scenarios, using both 'dynamic' and 'static' approaches (Imamura et al., 2012). Models couple expected evacuation performances with discrete or probabilistic tsunami floods to estimate mortality rates across evacuees. Examples of 'dynamic' evacuation models comprise, for instance, agent-based (Aguilar and Wijerathne, 2016; León et al., 2019; Makinoshima et al., 2016, 2018; Mas et al., 2015; Mostafizi et al., 2017; Taubenböck et al., 2009; Wang et al., 2016; Wang and Jia, 2020), cellular automata (e.g. Kitamura et al., 2020), and the 'evacuee generation model' of Dohi et al. (2016) (which includes the effect of external information on the evacuation behaviour). In turn, GIS-based, least-cost-distance (e.g. Fraser et al., 2014; Priest et al., 2016; Wood et al., 2018, 2020) and network approaches (Dewi, 2012; González-Riancho Calzada et al., 2013) are examples of 'static' evacuation models (whilst sometimes including variability in wave arrival times, population exposure scenarios, evacuation departure times, and travel speeds), which allow the identification of 'evacuation landscapes' (Wood et al., 2014).

As Goto and Nakasu (2018) point out, a quantitative analysis of the relationships between fatalities rates and geographical, built environment, and socio-psychological features can support the development of effective measures to reduce the loss of human lives. Moreover, if place-based models' findings can be generalised, this "will produce a tool for measuring areal vulnerability to future tsunamis and enable municipalities to prioritise the order of their countermeasures" (Goto and Nakasu, 2018, p.2). In the case of evacuation as a method for reducing disaster vulnerability, authors like Perry et al. (1981), Mohareb (2011), and Murray-Tuite and Wolshon (2013) provide analyses of geomorphological and socio-psychological aspects that determine the evacuees' behaviour during an emergency (e.g. selection of the escape routes, the required times for evacuation, human response, travel and waiting, and the role of the crowd influence). In this respect, for the case of tsunamis Makinoshima et al. (2020) deliver a comprehensive review of evacuation behaviours during 22 events since the Chilean tsunami of 1960, built around a framework with 3 stages of notifications: early, mid and late. They found out a range

of possible evacuation notifications across these stages, including ground shaking, official warnings, informal communications, and natural signs (e.g. unusual sea level changes, sighting the landing tsunami, hearing unusual sounds). In turn, these notifications can motivate a range of risk cognition and response activities, with sharp variations among different individuals and groups (determined by factors like previous knowledge or experience, culture, mental biases, and geographical location. These response activities include, for instance, collecting information, confirming the safety and gathering of family members, preparatory actions (e.g. packing emergency kits, collecting important goods). Following these activities, the actual evacuation begins, determined by a destination, a route of travel, and a means of evacuation (e.g. pedestrian or vehicular). Lastly, post-evacuation activities might include gathering additional information, contacting family or friends, or returning home (e.g. to pick up valuables or get the car).

While socio-psychological aspects can be critical determinants of evacuation, we will focus our research on some of the most relevant attributes of the geographical and built environments (capable of being quantitatively assessed through computer-based modelling) that could contribute to the success (or failure) of evacuation in the case of a tsunami. These characteristics include those related to the tsunami (maximum flood depth, and the estimated arrival time of this maximum depth), context (elevation, distance to the shoreline), the evacuation process (travel time, distance to the shelter, route length, pedestrian directness ratio), and the street network configuration (betweenness, closeness, straightness). In this respect, authors like Allan et al. (2013), Kubisch et al. (2020), Tumini et al. (2017), Villagra et al. (2014), and Villagra and Quintana (2017) underline the links between urban morphology/geospatial characteristics and evacuation. They point out how the former physically affects the latter and examine how behavioural aspects (e.g. the decision of evacuation, route selection or evacuation mode) relate to the environmental factors. In line with this, following Goto and Nakasu (2018), we aim at quantitatively assessing the relationship between the geographical and built environments' attributes and tsunami vulnerability (represented by the expected death ratio) as a first step towards the proposal of evidence-based countermeasures for risk reduction. For instance, as most evacuations take place in cities, planners and decision-makers could apply our recommendations for built environment changes and standards (aimed at increasing the number of evacuees that can reach safe areas) to guide the physical development retrofitting of tsunami-prone coastal communities around the world. This, with the final aim of enhancing pedestrian evacuation, saving lives, and therefore increasing resilience.

León et al. (2021b) deliver a modelling framework (including flood and agent-based evacuation) to examine the relationship between the evacuation potential and urban form characteristics of 67 urban samples from 12 case studies in Chile. In turn, they use the model's outcomes to develop a multivariate regressive analysis, which allows them to 'weight' the relative importance of each of the independent variables (i.e. the urban form characteristics) on the evacuation times. In this paper, we propose to enhance their approach with a greater emphasis on the description of real-world geographical and built environment's conditions that might influence tsunami evacuation. Therefore, while León et al. (2021b) set up a generic tsunami scenario where they test selected urban samples for flood and evacuation, we aim at developing a multi-case study approach that encompasses real-world-based large flood and evacuation models for seven coastal cities in Chile: Arica, Iquique, Coquimbo, La Serena, Viña del Mar, Valparaíso, and Talcahuano. Moreover, we focus our descriptive and

multivariate regressive analyses on the expected death ratios of these cities' exposed areas (as an indicator of human vulnerability to tsunamis) and how they can be affected by the geographical and built environment characteristics.

The rest of this paper is as follows. Section 2 describes the methodology, which comprises the selection of the seven examined Chilean cities and a description of two scaffolding cross-case research phases: a descriptive statistical analysis and a multivariate regressive analysis. Section 3 presents the results of our research, which we discuss in section 4. Lastly, section 5 delivers the study's main findings and proposes paths for future investigation.

## 2 Methodology

### 2.1 Case studies

Chile is one of the most tsunami-prone countries globally, with more than 100 tsunamis recorded since the 16th century, including 35 destructive events up to 2005 (Lagos and Gutiérrez, 2005), and recent disasters in 2010, 2014 and 2015. Moreover, researchers including Drápela et al. (2021), Klein et al. (2017), and Medina et al. (2021) have underlined the existence of extensive submarine areas in seismic locking along the central and northern coasts of Chile, capable of triggering large destructive tsunamis if major rupture earthquakes occur. Among the Chilean coastal cities, we selected seven case studies, distributed from north to south: Arica, Iquique, Coquimbo, La Serena, Viña del Mar, Valparaíso, and Talcahuano (see Fig. 1).

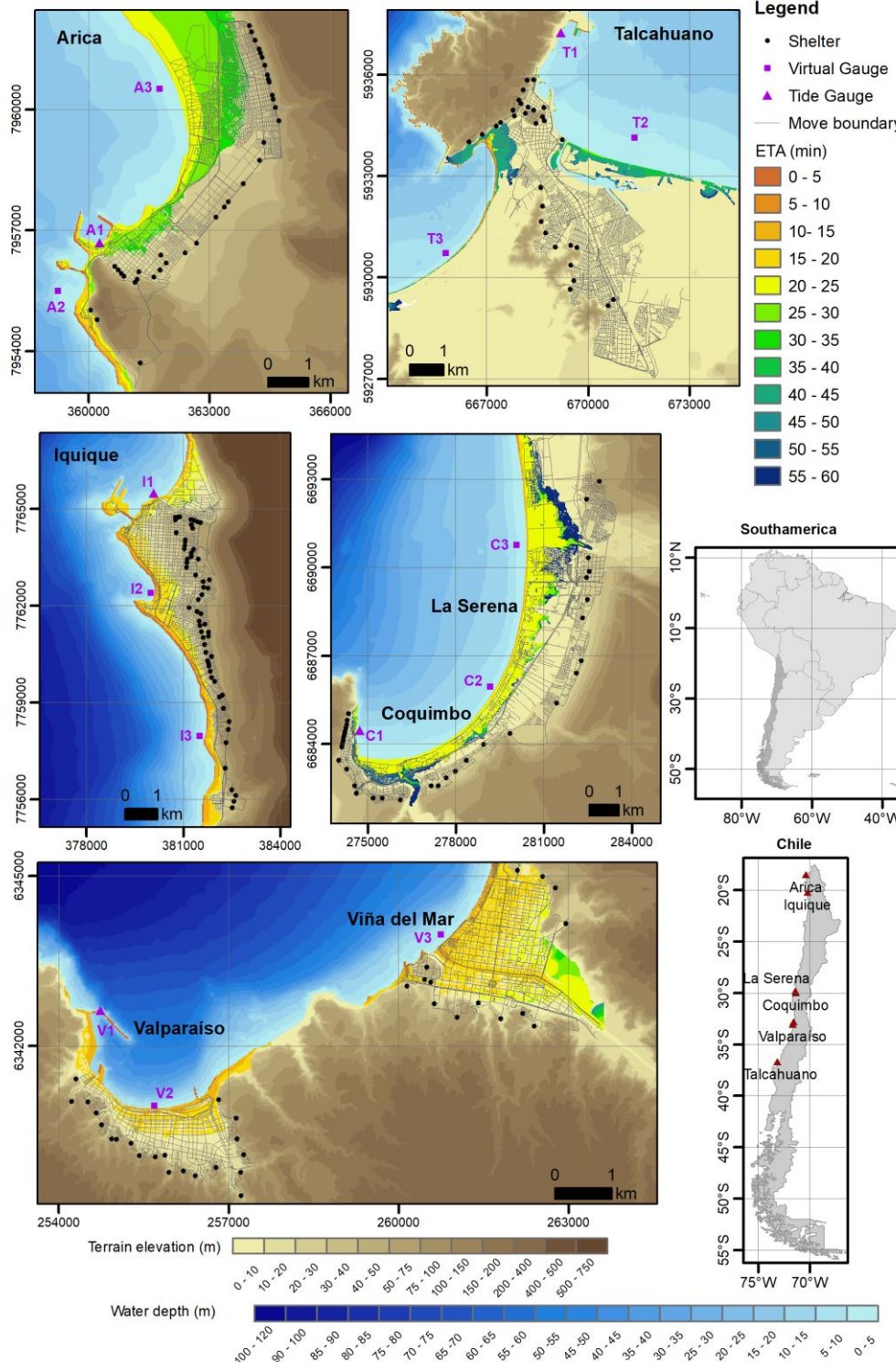

Fig. 1: Location, **topo-bathymetry** and **tsunami**-related features of the examined case studies in Chile.

According to the Chilean Bureau of Statistics (INE), these cities are among the top-20 in Chile with the most significant

ratios of exposed populations to tsunamis (INE, 2021). This information is based on census data and the official tsunami flood charts by SHOA (the Chilean Navy's agency aimed to provide technical elements, information and technical assistance to offer navigational safety in Chilean waters) (SHOA, 2012). Talcahuano, Iquique, and Arica occupy the first three places in the list, with 43.01%, 29.77%, and 23.44% of their populations living in floodable areas, respectively. In each of these cities we focused our analysis only on inhabited areas. Overall, the seven case studies gather roughly 240,000 exposed residents.

As seen in Table 1, historical records (since the 16th century) show that destructive tsunamis have repeatedly affected these cities.

Table 1: Attributes of the examined case studies in Chile.

| Case study | Location | Total population (census 2017) | Exposed resident population (CITSU) | Ratio of exposed resident population (%) | Years of recorded destructive tsunamis | Modelled population for evacuation (daytime scenario, departure time = 8 min) | Source of daytime population | Total number of 4x4 m cells | Number of 'lethal' 4x4 m cells |
|---|---|---|---|---|---|---|---|---|---|
| Arica | Northern Chile | 221,364 | 51,888 | 23.44 | 1604, 1868, 1877 (Lomnitz, 2004) | 81,420 | Call Detail Records (CDR) provided by Movistar (May 8, 2019, between 10:00 and 11:00) | 52,358 | 11,159 |
| Iquique | Northern Chile | 191,468 | 57,000 | 29.77 | 1604, 1868, 1877, 2014 (Catalán et al., 2015; Lomnitz, 2004) | 109,891 | Origin-destination study by SECTRA (2014) | 108,689 | 32,296 |
| La Serena | Northern Chile | 221,054 | 19,939 | 9.02 | 1849, 1922, 2015 (Aránguiz et al., 2016; Lomnitz, 2004) | 172,631 | Call Detail Records (CDR) provided by Movistar (January 19, 2019, | 211,451 | 20,844 |
| Coquimbo | Northern Chile | 227,730 | 6,240 | 2.74 | | | | | |

| | | | | | | | | |
|---|---|---|---|---|---|---|---|---|
| | | | | | | between 22:00 and 24:00) | | |
| Viña del Mar | Central Chile | 334,248 | 35,096 | 10.5 | 1730, 1822 (Carvajal et al., 2017; Lomnitz, 2004) | 62,519 | Origin-destination study by SECTRA (2016) | 37,859 | 20,592 |
| Valparaíso | Central Chile | 296,655 | 4,450 | 1.5 | 1730, 1822 (Carvajal et al., 2017; Lomnitz, 2004) | 32,492 | Origin-destination study by SECTRA (2016) | 16,063 | 7,038 |
| Talcahuano | Southern Chile | 151,749 | 65,267 | 43.01 | 1570, 1657, 1751, 1835, 1868, 2010 (Fritz et al., 2011; Lomnitz, 2004) | 34,996 | Call Detail Records (CDR) provided by Movistar (May 8, 2019, between 10:00 and 11:00) | 103,671 | 774 |

## 2.2 Descriptive analysis

This phase aimed to develop a thorough description of the current geographical and built environment conditions that might influence the outcome of tsunami evacuations in each of the case studies and, second, to integrate those results through GIS spatial post-processing based on 4x4 m cells as the basic units of study.

### 2.2.1 Tsunami inundation and evacuation models

We developed coupled tsunami inundation and evacuation models for each case study, using the methodologies extensively described in León et al. (2019) and León et al. (2020). First, we worked out flood simulations according to the worst-case feasible seismic scenario (i.e. a high consequence event of a relatively small likelihood (Løvholt et al., 2014)) for each city. To do this, we used the Storm Surge and Tsunami Simulator in Oceans and Coastal Areas (STOC), specifically the Multi-layered Static Dynamics Model (STOC-ML) (Tomita et al., 2006). We used seismic models by Carvajal et al. (2017) and Fujii and Satake (2012), and for the Iquique scenario we calculated the seismic parameters, including length, width, and slip, according to the scaling law by Papazachos et al. (2004) (see Table 2 and Fig. 2). The input data for the simulations included bathymetry, coastline, topography, and elevation data, compiled from various sources including SHOA, local governments

and GEBCO (www.gebco.net). Each case study's simulation used five nested grids for numerical analysis, with spatial resolutions of 1,536, 256, 32, 8 and 4 m, respectively. Tsunamis were simulated for 45 minutes (comprising its development from the occurrence of the earthquake to the maximum inland penetration, i.e. the inundation line or run-up). While sea level anomalies may last for several hours after the earthquake, preliminary tests showed that this threshold was sufficient to encompass the total evacuation time of each case study. The model used a time step of 0.1 s, also recording time series, the inundation depth (every 10 min), the maximum inundation depth, and the estimated arrival time of this maximum depth.

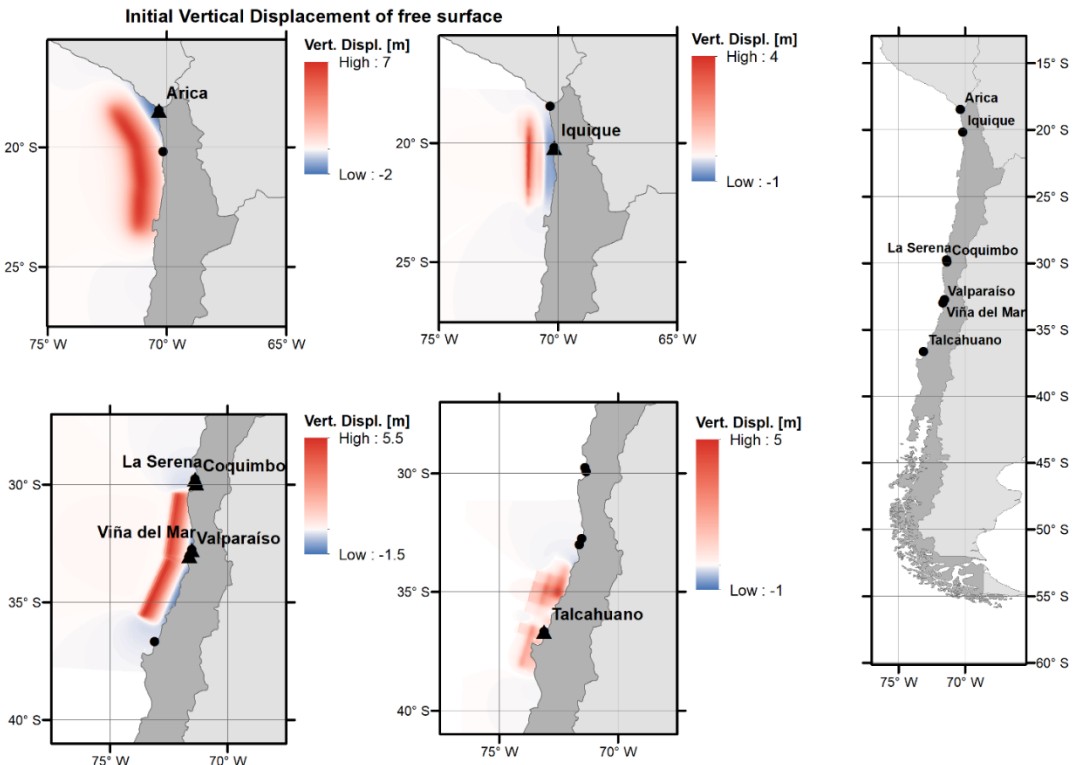

Fig. 2: Seismic scenarios used for the examined case studies.

Table 2: Seismic parameters of the examined case studies.

| Case study | Mw | Total length [km] | Total width [km] | Slip [m] | Source | Description |
|---|---|---|---|---|---|---|
| Arica | 9.0 | 600 | 150 | Uniform slip 17.0 [m] | Own | Large earthquake and tsunami |
| Iquique | 8.5-8.7 | 500 | 160 | Variable slip with peak of 10.0 [m] | Matías Carvajal | As a result of the accumulated slip since 1877 |

| Coquimbo, La Serena, Valparaíso and Viña del Mar | 9.1-9.3 | 600 | 180 | Variable slip with peak of 19.7 [m] | Carvajal et al. (2017) | The 1730 Valparaíso Earthquake |
|---|---|---|---|---|---|---|
| Talcahuano | 8.8 | See the Fujii and Satake (2012) model for the Maule 2010 earthquake, developed from tsunami and coastal geodetic data | | | | |

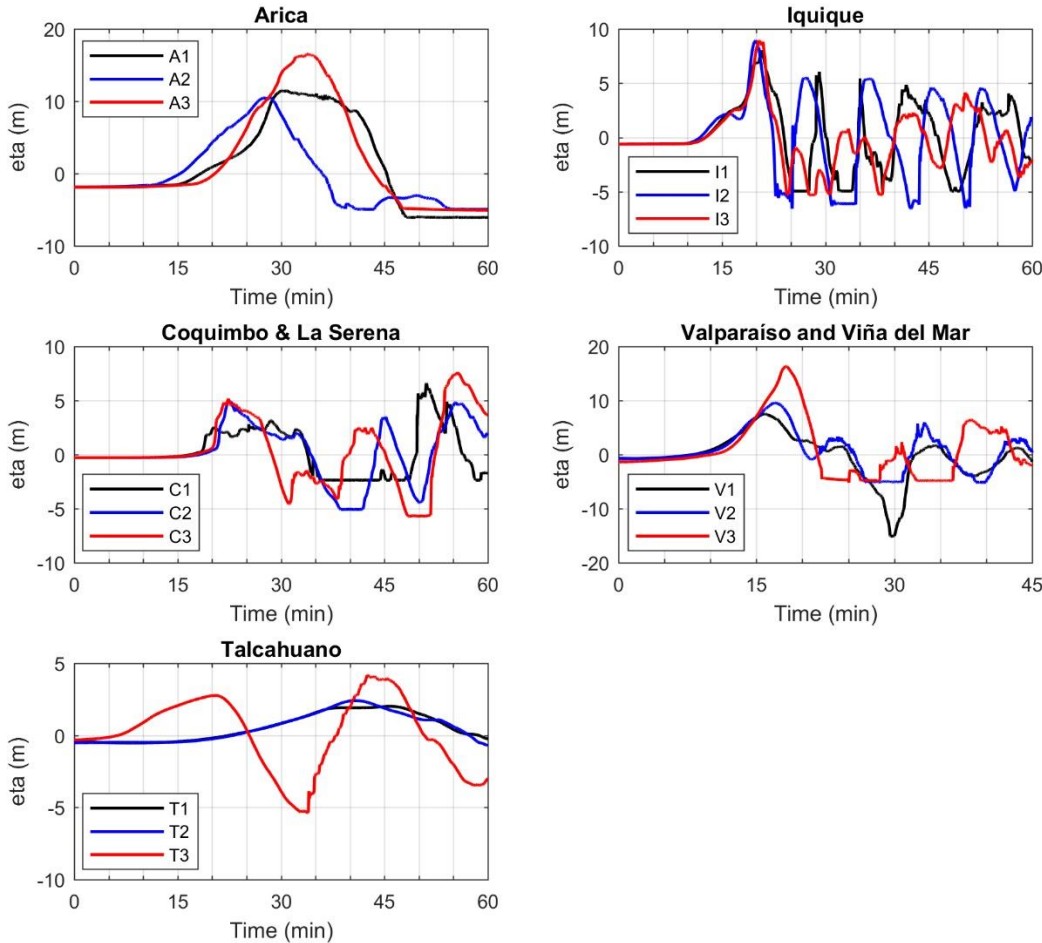

**Fig. 3: Tsunami time series for the examined case studies, as measured by virtual gauges (see their locations in Fig. 1).**

We also developed agent-based evacuation simulations (see section 1) for each case study, using an enhanced version of the PARI-AGENT model (Arikawa, 2015). We modified this source code to assess the impact of the slope on the evacuees' speed, to include a Rayleigh probabilistic distribution of departure times for evacuees, and to assign a range of age-based evacuation speeds across the agents (see León et al. (2020)). Recently, this model was validated using real-world data of

1,966 pupils from four K-12 schools in Valparaíso and Viña del Mar, Chile, collected during an evacuation drill held in September 2019 (León et al., 2021a).

Agent-based models are bottom-up computer simulations where individual disaggregated elements (the agents, which in our model correspond to evacuees) are modelled as autonomous decision-making entities that follow simple rules, which are iteratively performed within a set time threshold, usually including stochastic features. The agents' interactions (also with their environment) lead to emergent phenomena, which is helpful for the examination of complex, real-life events like mass evacuation. As we adopted a cross-case study perspective to examine extensive urban areas (534,881 agents overall), with a

focus on the geomorphological conditions affecting the evacuation, we formulated our agent-based model under a macroscopic perspective, disregarding microscopic interactions among agents (Makinoshima et al., 2018). Rather, we examined the overall outcomes of the evacuation processes (i.e. number of saved, moving and dead evacuees, for each time step), determined by the environmental conditions and the moving crowd (see below). To develop the agent-based models for each case study, we had to follow these steps. First, we included the inundation parameters obtained from the STOC-ML

analyses. Second, we determined the evacuation territories of each case study (henceforth denominated 'move boundary', see Fig. 1), comprising the streets and open spaces connecting the coastline with the safe assembly areas (shelters) as defined by the evacuation plans from ONEMI, the Chilean Emergency Management Agency (available at https://geoportalonemi.maps.arcgis.com/apps/webappviewer/index.html?id=5062b40cc3e347c8b11fd8b20a639a88). For their spatial definition, these move boundaries used the smallest nested grid from the inundation model (with 4x4 m cells).

We obtained their specific configuration through its intersection with the street network obtained from OpenStreetMap (https://www.openstreetmap.org/) and post-processed in ArcMap 10.4.1 (see Fig. 1). Third, we had to establish worst-case daytime population distributions (different from census data), reflecting that most of the examined zones comprise downtown, CBD, or touristic areas that significantly increase their populations during daytime due to commuting and visiting (see Table 1). In the case of Iquique, Viña del Mar, and Valparaíso, we obtained daytime populations from previous

origin-destination studies conducted by the Chilean Ministry of Transportation (SECTRA, 2014, 2016). For Arica, Coquimbo, La Serena, and Talcahuano, in turn, we used extrapolations of mobile CDR (call detail records) databases provided by one of the largest telecom companies in Chile (with a market share of roughly 28%). In the case of Arica and Talcahuano, we used the morning peak time (10:00 to 11:00) of a random weekday (May 8, 2019) as the worst-case daytime scenario. For Coquimbo and La Serena, popular summer touristic destinations with vibrant nightlife along their coastlines,

we used Saturday, January 19, 2019, between 22:00 and 24:00. The PARI-AGENT code randomly distributed these populations across the move boundaries within each case study (locating one or more agents on each 4x4 m cell). Fourth, we established the agents' performance parameters, including: (1) the impact of the slope on the evacuees' speed, according to Tobler's exponential hiking function (Tobler, 1993); (2) a Rayleigh probabilistic distribution of departure times for evacuees (see Mas et al., 2012a), with a mean of the distribution (the μ factor) equal to 8 min, which corresponds to the average time

that ONEMI takes to release an evacuation warning); (3) an evacuation speed for each agent, according to its age (Buchmueller and Weidmann, 2006), probabilistically defined based the case studies' population pyramids from the 2017

Census (INE, 2018); (4) a random-walk parameter that introduces an aleatory fluctuation up to 10º on the evacuation direction; and (5) a crowd potential parameter that makes the agent tend to follow the direction in which other evacuees are moving, stochastically assigned (with a probability of 0.5). Fifth, we executed the simulation, in which the code initially computes the optimal route for each agent, according to its initial position and closest shelter, using the A* algorithm, frequently applied in evacuation studies (e.g. Mas et al., 2012a; Takabatake et al., 2020; Wang and Jia, 2021). Then, it calculates every agent's position at each time step (1 s), based on its departure time and velocity (which could be modified by the slope, random-walk and crowd parameters). The code compares this new position with the water height at that moment (obtained from the inundation file) and updates the agent's status: (1) moving (i.e., alive), (2) dead (i.e., reached by the water), or (3) escaped (i.e., alive in the shelter). This process continues for 45 min. and then the computation stops.

As the model included stochastic parameters (the initial positions of the agents, their walking speeds and departure times, and the random-walk factor), we carried out ten simulation rounds for each case study, intending to achieve a 95% confidence interval with a margin of error <1% in the average values of the number of escaped, moving and dead evacuees after 45 min. The model also recorded each agent's travel time and evacuation route (for every iteration).

### 2.2.2 Street network configuration model

According to Fakhrurrazi and Van Nes (2012), an appropriate street network configuration can increase the evacuees' chances of successfully evacuating in case of a tsunami. The suitability of a street network for evacuation depends on factors like its accessibility, variety of route options and the possibility of short, direct trips (Dill, 2004; Handy et al., 2003). While a range of metrics has been proposed to examine these characteristics (Sharifi, 2019), we will focus our analysis on centrality indicators, which can be used "to measure the degree of importance of specific nodes/links in a street network" (Sharifi, 2019, p.174), based on how central the locations are compared to the rest of the urban layout (Porta et al., 2006). Moreover, centrality is a good predictor of everyday human movement (Sasabe et al., 2020; Turner, 2007), and authors like Mohareb (2011) and Marín Maureira and Karimi (2017) point out that evacuees tend to choose well-known paths instead of the designated ones.

We examined the move boundaries (described in section 2.2.1 above) from each case study with the Urban Network Analysis Toolkit for ArcMap (UNA) (Sevtsuk et al., 2013). For each street segment belonging to the input network, we analysed three centrality metrics: (1) betweenness, (2) straightness, and (3) closeness. These can be defined, respectively, as (Sevtsuk et al., 2013; Sharifi, 2019): (1) the fraction of shortest paths between all pairs of destinations in the street network that pass through an examined street segment; (2) the extent to which the shortest paths from a segment of interest to all the other segments in the street network resemble straight Euclidean paths; and (3) an indication of how close a street segment is to all other street segments in the network. To compare the street network's components, the toolkit normalises the outcomes according to the total number of segments in the network. The Urban Network Analysis Toolkit delivers its outputs as a new GIS vector shapefile, with the input street network including these metrics.

### 2.2.3 Context-determined evacuation metrics

Each discrete location belonging to the examined areas of each case study (represented in our model by a 4x4 m cell) has a set of evacuation metrics determined by its existing spatial relationships to the geographical and built contexts. We examined these metrics using ArcMap 10.4.1 and the same data sources mentioned in sections 2.2.3 and 2.2.4 above. These indicators include (1) elevation; (2) sea distance, i.e. the straight-line distance between the cell's centre and its closest shoreline point; (3) distance to shelter, i.e. the straight-line distance between the cell's centre and the closest safe assembly area; and (4)

pedestrian directness ratio (PDR) (also termed "Pedestrian Route Directness, (PRD)" by Dill (2004)), which is the ratio of the 'real-world' route distance (as determined by the street network) and the straight-line distance connecting the cell's centre and its closest safe assembly area. In this respect, Hillier and Iida (2005) and Hillier (2009) underline that the strongest movement predictor is the least angle change along the routes. Therefore, networks including fewer direction changes (i.e., lower PDR values) might improve the evacuees' wayfinding performance. Wayfinding is "the process of

determining and following a path or route between an origin and a destination" (Allen, 1999, p.6).

### 2.2.4 Spatial post-processing

In this research phase, we aimed to integrate the previous sections' outcomes into a descriptive spatial analysis that could also serve as the basis for the subsequent multivariate regressive study. We carried out this integration with the aid of the ArcMap 10.4.1 software. Our canvas included the move boundaries described in section 2.2.1 above for each case study,

which comprised a network of streets and open spaces represented by raster files with 4x4 m cells. Each of these cells corresponded to a specific location in the evacuation landscape, for which all the calculated metrics had to be spatialised. First, as the inundation and agent-based models used the same base raster, the former's results did not need to be post-processed. Second, the data from each case study's evacuation model included a range of at least ten different groups of agents' initial locations (each with an associated final status: moving, dead, or escaped). Due to our purpose of examining

vulnerability, we aimed at quantifying, for each cell, its death ratio (i.e. the percentage of dead agents that began their journey from it, comprising all the model's iterations). To do this, we used the ArcMap's Spatial Join Tool, which joined the attributes from the source feature (i.e. the initial locations of agents, all merged in a single shapefile) to the target feature (i.e. the raster-based street network). Third, for the case of the street configuration model, we applied the same Spatial Join tool to cast the properties from the outcoming street network into the base raster. Lastly, as we also calculated the context-

determined evacuation metrics on the base raster, their results did not need to be post-processed, either.

Our analysis, comprising all the case studies, included 530,091 cells, each of them containing the following data fields: (1) death ratio, (2) maximum, (3) minimum and (4) mean travel time, (5) sea distance, (6) elevation, (7) total route length, (8) shelter distance, (9) estimated arrival time (ETA) of the maximum flood, (10) maximum flood, (11) closeness, (12) betweenness and (13) straightness (and their three normalised values), (17) pedestrian directness ratio (PDR), and the UTM

latitude and longitude coordinates (18 and 19). Assuming that the death ratio was our dependent variable, we ran a

correlation test to prevent the correlation between the other predictor (independent) variables and therefore avoid collinearity problems in the regressive model (see Fig. 4). This test demonstrated that seven of these variables (numbers 2, 3, 8, 12, 14, 15 and 16 above) should not be included in the analysis, as they were correlated at $|r| > 0.7$ (Dormann et al., 2013). Neither we included the UTM coordinates, as they are defined according to global reference systems and no to local conditions.

Table 1 shows the number of examined cells for each case study.

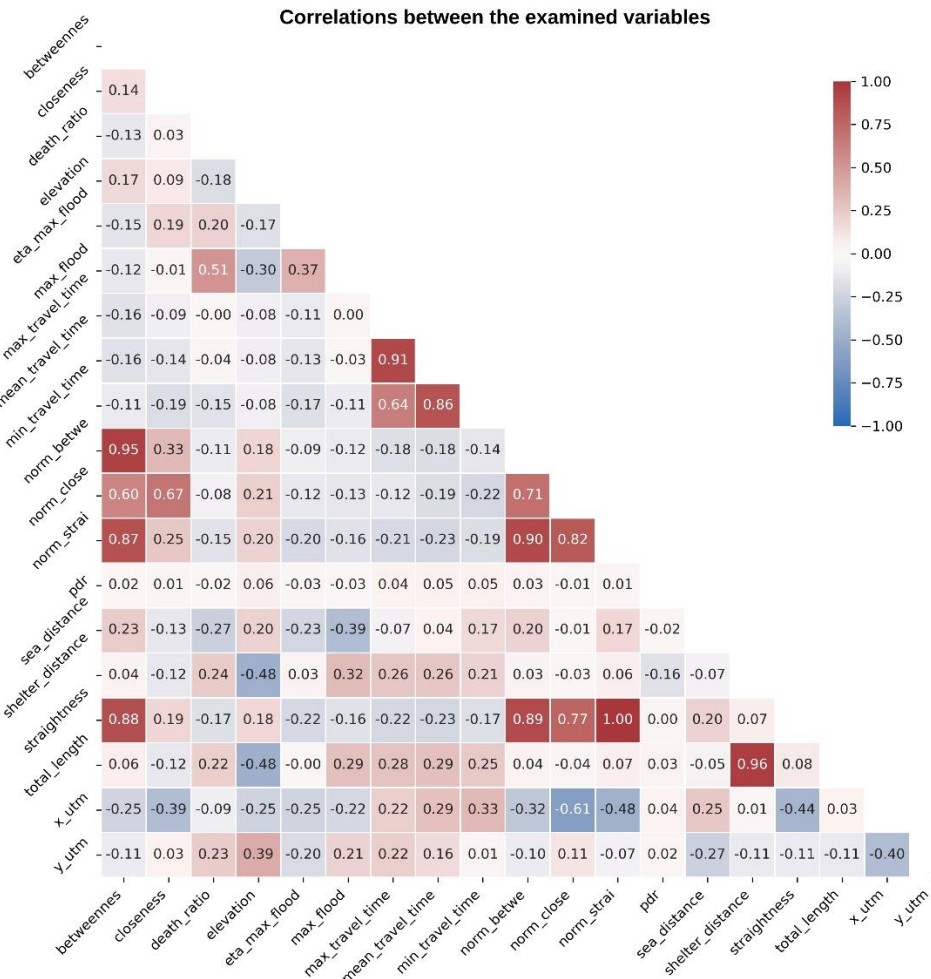

**Fig. 4: Correlations between the examined variables.**

## 2.3 Multivariate regressive analysis

For each case study, the result from the spatial post-processing was a raster shapefile representing the evacuation territory,

where each cell included the 19 data fields mentioned in section 2.2.4. The objective of our regressive analysis was to test, for each of the 530,091 cells, the death ratio (the dependent variable) against the other 9 selected independent variables, which represent characteristics of the geography and built environment. In this way, we could examine how much each of

these characteristics contributes to the expected death ratios. To do this, we developed a multivariate regressive analysis using a random forest methodology, which combines a multitude of simple decision trees (Breiman, 2001). Tree-based methods for regression and classification stratify or segment the predictor space into several simple regions. To predict a given observation, we typically use the mean or the mode response value for the training observations in the region it belongs. Since the splitting rules used to segment the predictor space can be summarised in a tree, these approaches are known as decision tree methods (James et al., 2013). Random forest is an ensemble method that combines many simple decision trees models to obtain a single and potentially powerful model. Each tree takes random samples of the observations and performs split steps using a subset of features, thereby decorrelating the trees and leading to a more thorough exploration of model space.

We applied a K-Fold cross-validation method to assess this model's outcomes (Mosteller and Tukey, 1968). Following this method, we randomly split all the input data (comprising the 9 independent and 1 dependent variables) into five equal-size packages (folds). In four of them (the training packages), we applied our regressive random forest model to internally predict the values of the death ratio, according to the independent variables; this was the 'training' test. We repeat this process for the 'external' fifth package (the testing one) on the 'testing' test. Then, we calculated the coefficient of determination ($R^2$) to assess the strength of the relationship between the predicted and actual dependent in the 'training' and 'testing' packages. After that, we tried the other four combinations of training and testing packages to obtain further $R^2$ scores. Then, we executed other five random splits of the input data, leading to overall 30 repetitions of the procedure. Overall, our mean $R^2$ scores were 0.9101 (SD=0.0021) and 0.8607 (SD=0.0022) for the 'training' and 'testing' analyses, respectively, which underline our model's goodness-of-fit. Lastly, to enhance the interpretation of the model's results, we used SHAP values (Lundberg and Lee, 2017).

SHAP (SHapley Additive exPlanations) values is a unified framework for explaining model predictions, motivated by the idea that model interpretability is as important as model accuracy, since some modern models act as black boxes due to their complexity. It has three significant advantages over other explanatory approaches: (1) it considers that interpreting a prediction model is a model itself, commonly named as an explanation model; (2) game theory results guarantee a unique solution; and (3) the method is better aligned with human intuition. SHAP values is a unique unified measure of feature importance since it meets three desired properties: (i) local accuracy (approximating the original model); (ii) missingness (a missing feature in the original input have no impact); and (iii) consistency (if a model changes, then the attributes of the inputs should be updated as well). Classic Shapley regression values examine feature importance for linear models in the presence of multicollinearity. To do this, SHAP retrains the model on all feature subsets, assigning an importance value to each feature that represents the effect of including that feature on the model prediction: a model is trained with a particular feature present, and another model is trained with the feature withheld, and then predictions from the two models are compared (Lundberg and Lee, 2017). SHAP values are the Shapley values of a conditional expectation function of the original model. The exact computation of SHAP values is challenging. However, combining insights from current additive

feature attribution methods makes it possible to approximate them, leading to good computational efficiency. Since this method is, essentially, a sum of the contributions of each feature, which is consistent with human intuition.

To develop our statistical analysis, we used an ad-hoc Python model, comprising the data analysis libraries NumPy (Berg et al., 2020) (https://numpy.org/), pandas (McKinney et al., 2020) (https://pandas.pydata.org/), and SHAP (Lundberg, 2020) (https://github.com/slundberg/shap).

## 3 Results

### 3.1 Death ratios and the natural and built environment's characteristics

Table 3 summarises our descriptive analysis comprising 530,091 cells belonging to the case studies' move boundaries. This table arranges the data in 11 intervals according to growing (in 10% steps) death ratio thresholds. For each of these intervals, we include the mean and standard deviation values of 9 independent variables (mean travel time, sea distance, elevation, total route length, estimated arrival time (ETA) of the maximum flood depth, maximum flood depth, closeness, straightness, and pedestrian directness ratio (PDR)). Figure 5, in turn, shows scatterplots (summarizing all the examined cells with positive death ratios) of the death ratio compared with each of the nine independent variables, plus one extra chart that shows the death ratio distribution across the case studies. To enhance readability, we post-processed the 530,091 scattered records into a 10 x 10 grid, where each square's colour depth represents the percentage of all the data comprised by it.

Our analysis shows that 92,703 of 530,091 cells (17.49%) have at least one 'dead' agent (evacuee) across the simulations. Moreover, Fig. 5 shows that Arica, Iquique, Valparaíso, and Viña del Mar have cells where the death ratio reaches up to 1, i.e. every agent departing from them is caught by the modelled tsunamis. The analysis also shows that some of the geographical and built environment's attributes have clear spatial relationships with death ratios. In this respect, it is useful to compare their average values for two different death ratio thresholds: 0 and >0.0 (i.e. without and with 'dead' agents in the cells), to highlight their differences. For instance, cells with positive death ratios have an average elevation of 5.39 m.a.s.l., i.e. 39.03% of the average value of the 'safe' cells (13.82 m.a.s.l.). In line with this, Fig.5 shows that the maximum values of elevation for 'deadly' cells stay below 20 m.a.s.l.. In the case of the distance to the sea, the ratio between the average values of 'deadly' and 'safe' cells is 0.25 (296.57 and 1,185.38 meters, respectively), with maximum levels smaller than 1,750 metres (according to Fig. 5). For the maximum flood depth attribute, the ratio is 0.13 (average values of 0.611 and 4.67 meters for the 'safe' and 'deadly' cells, respectively), with no cells above 15 metres. A similarly steep difference occurs in the case of the ETA of the maximum flood, with a ratio of 0.13 due to an average value of 49.16 seconds for those cells with a death ratio = 0, and of 364.76 seconds for cells with a death ratio >0.0. In the case of the variables related to evacuation and urban form parameters (mean travel time, total route length, closeness, straightness, and pedestrian directness ratio (PDR)), the differences between the average values of 'safe' and 'deadly' cells are also pronounced in some cases (0.57 for straightness and 0.64 for both total route length and closeness), while mild in others (0.97 and 0.93 for PDR and mean travel time, respectively).

To assess the dispersion of results shown by Fig.5, it is useful to compare the coefficients of variation of each data field, in the case of the average values of two different death ratio thresholds (0 and >0.0). According to Abdi (2010), the coefficient of variation allows the comparison of data distributions with different units. It is defined as the standard deviation of a series of numbers, divided by the mean of this series of numbers. Along these lines, in the case of the 'safe' cells (death ratio = 0), the examined data fields (mean travel time, sea distance, elevation, total route length, estimated arrival time (ETA) of the maximum flood, maximum flood depth, closeness, straightness, and pedestrian directness ratio (PDR)) show coefficients of variation of 0.96, 0.74, 0.89, 0.74, 4.95, 2.98, 1.26, 1.22, and 0.41, respectively. On the other hand, for the 'deadly' cells (death ratio >0.0) and the same variables, the coefficients of variation are 0.54, 0.95, 0.55, 0.47, 1.47, 0.67, 1.41, 1.62, and 0.35, respectively. These values show that, in the case of cells with a death ratio = 0, the dispersion of results is in a similar range, except for two variables: ETA of the maximum flood, and maximum flood depth. In the case of death ratio >0.0, variation among results is more limited, with three highlighted variables: estimated arrival time (ETA), closeness, and straightness.

Table 3: Death ratio thresholds and mean and standard deviation of the independent variables.

| Death ratio thresholds | Mean travel_time (sec.) | | Sea distance (m.) | | Elevation (m.a.s.l.) | | Total route length (m.) | | Estimated arrival time (ETA) of the maximum flood (sec.) | | Maximum flood depth (m.) | | Closeness | | Straightness | | Pedestrian directness ratio | | Number of examined cells |
|---|---|---|---|---|---|---|---|---|---|---|---|---|---|---|---|---|---|---|---|
| | Mean | S.D. | Mean | S.D. | Mean | S.D. | Mean | S.D. | Mean | S.D. | Mean | S.D. | Mean | S.D. | Mean | S.D. | Mean | S.D. | |
| 0 | 922.60 | 885.47 | 1185.38 | 879.00 | 13.82 | 12.34 | 910.11 | 672.28 | 49.16 | 243.40 | 0.61 | 1.82 | 9.88632E-07 | 1.24947E-06 | 36494.23 | 44508.04 | 1.40 | 0.580 | 437,388 |
| >0.0 | 858.13 | 460.51 | 296.57 | 282.07 | 5.39 | 2.98 | 1419.89 | 670.28 | 364.76 | 537.59 | 4.67 | 3.14 | 1.53525E-06 | 2.17001E-06 | 20874.97 | 33857.30 | 1.36 | 0.48 | 92,703 |
| >0.1 | 839.00 | 408.59 | 295.93 | 303.76 | 5.61 | 3.22 | 1452.12 | 643.17 | 354.02 | 524.79 | 5.13 | 3.34 | 1.42721E-06 | 2.16893E-06 | 15255.32 | 27795.81 | 1.37 | 0.49 | 65,353 |
| >0.2 | 832.36 | 366.71 | 307.76 | 324.29 | 5.89 | 3.40 | 1464.85 | 616.99 | 316.78 | 502.94 | 5.34 | 3.54 | 1.2593E-06 | 1.96081E-06 | 11463.28 | 21170.88 | 1.36 | 0.46 | 50,383 |
| >0.3 | 827.92 | 345.75 | 312.69 | 338.20 | 5.99 | 3.49 | 1478.09 | 608.92 | 296.55 | 490.50 | 5.50 | 3.67 | 1.15586E-06 | 1.80679E-06 | 9444.16 | 15931.57 | 1.35 | 0.45 | 42,245 |
| >0.4 | 820.13 | 332.13 | 310.76 | 343.27 | 5.99 | 3.50 | 1492.13 | 607.07 | 281.72 | 480.41 | 5.59 | 3.77 | 1.10805E-06 | 1.70410E-06 | 8388.61 | 12161.42 | 1.35 | 0.45 | 35,542 |
| >0.5 | 799.21 | 315.14 | 295.33 | 336.20 | 5.87 | 3.43 | 1508.79 | 610.24 | 282.44 | 481.27 | 5.66 | 3.82 | 1.08333E-06 | 1.63073E-06 | 7668.96 | 9223.47 | 1.34 | 0.45 | 28,329 |
| >0.6 | 786.67 | 313.91 | 299.90 | 352.55 | 5.88 | 3.53 | 1526.85 | 622.89 | 282.08 | 481.19 | 5.81 | 3.90 | 1.03059E-06 | 1.59139E-06 | 7337.72 | 8624.27 | 1.34 | 0.45 | 23,536 |
| >0.7 | 776.61 | 312.97 | 300.71 | 361.24 | 5.87 | 3.56 | 1530.47 | 627.62 | 268.94 | 474.22 | 5.82 | 3.97 | 1.00919E-06 | 1.50761E-06 | 7332.52 | 8225.81 | 1.33 | 0.42 | 19,888 |
| >0.8 | 774.60 | 318.28 | 310.38 | 376.05 | 5.91 | 3.63 | 1526.73 | 624.86 | 247.92 | 461.77 | 5.80 | 4.07 | 9.83519E-07 | 1.36354E-06 | 7420.44 | 7888.60 | 1.33 | 0.40 | 16,754 |
| >0.9 | 785.42 | 324.63 | 326.40 | 390.21 | 6.02 | 3.75 | 1499.34 | 624.41 | 216.67 | 440.36 | 5.70 | 4.15 | 9.28422E-07 | 1.04837E-06 | 7686.08 | 7969.91 | 1.3 | 0.38 | 14,753 |
| 1 | 797.02 | 325.87 | 337.88 | 398.07 | 6.09 | 3.83 | 1476.24 | 623.87 | 191.53 | 420.78 | 5.58 | 4.16 | 8.93807E-07 | 7.53117E-07 | 7945.43 | 8116.71 | 1.3 | 0.39 | 13,825 |

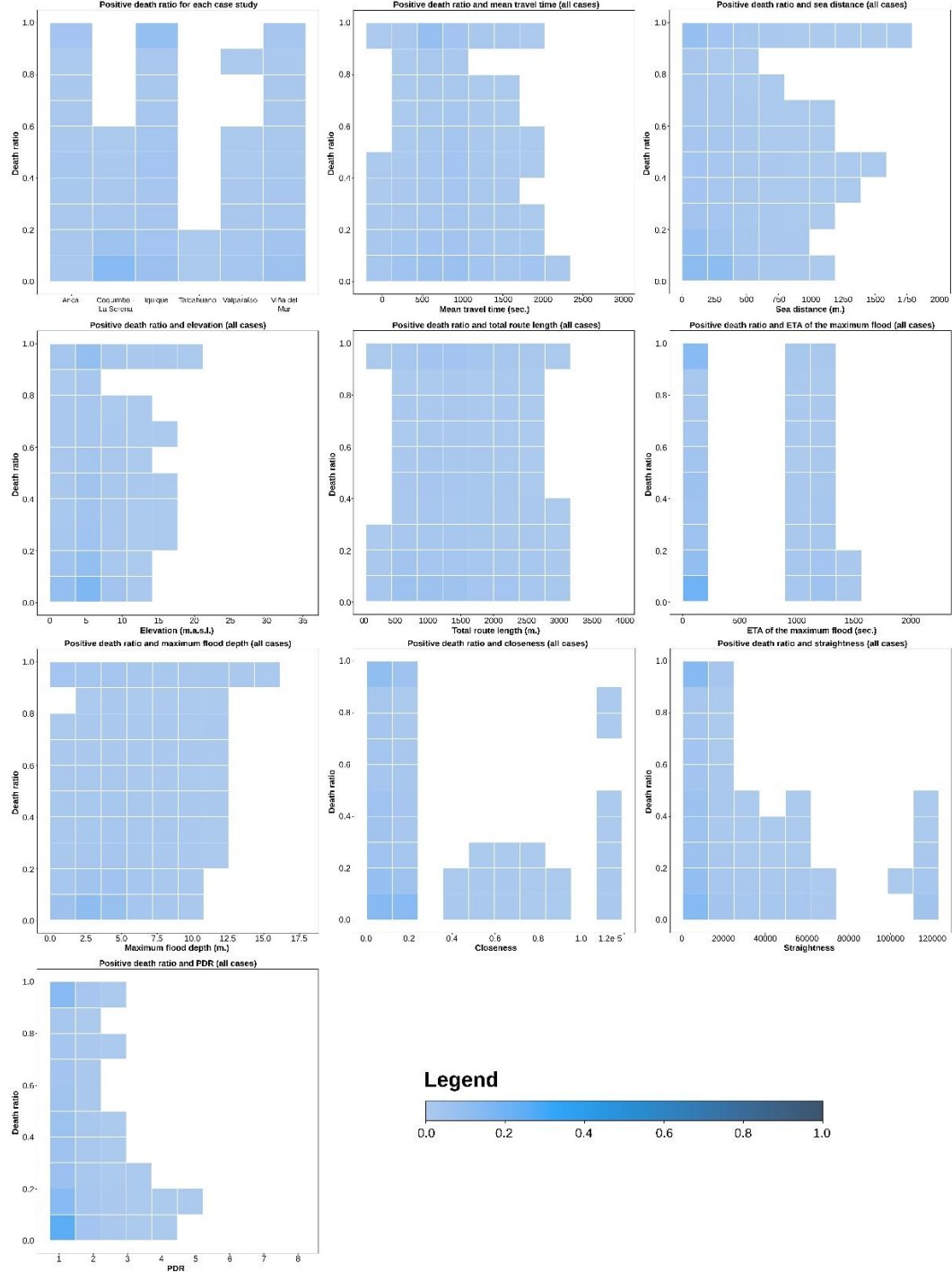

Fig. 5: Data scatterplots showing the distribution of the death ratios, in comparison to the nine examined independent variables. The prevalence of different death ratios across the case studies is also included (top left image).

## 3.2 Multivariate regressive analysis

Figures 6 and 7 show the results of the SHAP values analysis for the Random Forest model's outcomes. Figure 6 shows, for every independent variable and all the examined cells, the amount of the former's contribution (either positive or negative) to the predicted death ratio (compared to the average prediction across all the cells). Red dots mean higher values of the independent variable, while blue ones imply the opposite. Figure 7 processes this data to display, for each independent variable, the average absolute contribution to the predicted death ratio. These results show that the most important feature in predicting death ratios is the maximum flood depth, followed by the straightness, the total route length, and the mean travel time. On average, the maximum flood depth can vary the death ratio up to 0.08 points, more than twice the impact of the straightness (0.037). In turn, this value is higher than those of the total route length (0.032) and the mean travel time (0.024).

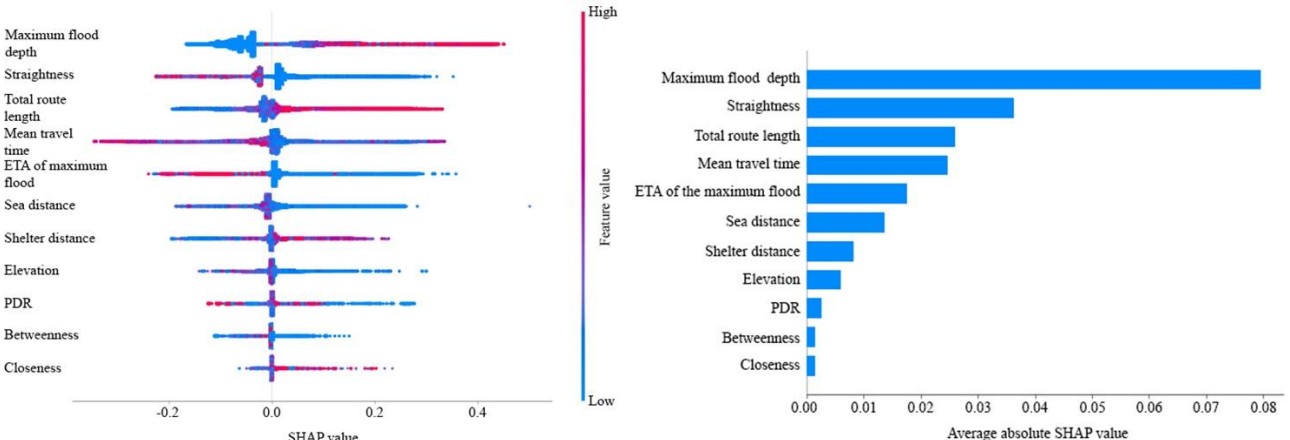

**Fig. 6: SHAP values of the independent variables**     **Fig. 7: Average absolute SHAP values of the independent variables**

## 4 Discussion

Our descriptive analysis included 530,091 cells. Of these, 92,703 (17.49%) have a death ratio >0.0 (i.e. at least one agent from any of the model's run, who started its evacuation from one of them, was caught by the tsunami). In turn, 13,825 cells (2.61%) have a death ratio=1, which means that the waters reach every agent departing from them before they arrive at a safe assembly area. As shown by Table 1 and Fig. 5, the rate of cells with elevated death ratios is unevenly distributed across the case studies. Cities like Viña del Mar, Valparaíso, and Iquique show large percentages of cells susceptible to having dead evacuees (54.78%, 43.81%, and 29.71%, respectively). On the contrary, Talcahuano has only 0.74% of its cells on this condition. As we can see in the maps include in Fig. 1, while the first three cities gather considerable urban development and residential populations on exposed locations right next to the coastline, most of the last one's territory is roughly 1.0 to 1.5 km from the coast, from whom large, marshy areas separate it.

The death ratio thresholds included in Table 3 and the results in Fig. 6 and Fig. 7 allow appraising, for the examined case studies, how each independent variable relates to the possibility of death in case of a tsunami, and how these variables change between 'deadly' and 'safe' locations. First, the data shows that three of the four of most important predictor variables (the maximum flood depth, straightness, and total route length, whose impacts are on average 5.1 times larger than the other eight examined variables) have significant differences between their average values for the 'deadly' and 'safe' cells (ratios of 0.13, 0.57, 0.64, respectively). Second, some of the variables exhibit what we might call an expected behaviour: the probability (for an agent) of 'dying' because of a tsunami increases if the departing cell has comparatively lower elevation or shorter distance to the sea. In a similar manner, a higher maximum flood depth also increases the cells' death ratios. Third, three variables show somewhat counterintuitive results: the death ratio increases when the mean travel time reduces, and when the estimated arrival time (ETA) of the maximum flood, and the closeness, grow. In the case of the first of these independent variables, the results are likely influenced by the fact that the evacuees departed from 'lethal' cells have comparatively shorter (actual) evacuation times, as the tsunami soon reaches them and cannot complete their evacuation paths. In the case of the street network's closeness (which is a measure of how close a cell is to all other cells within the 'evacuation territory'), one may expect that more compact street networks should lead to shorter evacuation routes (and times) and, therefore, less 'dead' evacuees. Nevertheless, according to our results, it would be possible that these smaller networks are faster to flood by the incoming tsunami. Lastly, in the case of the ETA, it is essential to underline that the estimated arrival time of the maximum flood is not necessarily the same as the onset time of the first tsunami front. In our model, the latter can have much more impact on the evacuees' survival rates.

While tsunamis and their related evacuation potentials are highly context-dependent, our cross-case results could serve to identify and appraise other tsunami-vulnerable areas in Chile. Moreover, they highlight possible spatial planning guidelines that could be applied to develop new urban regions into exposed territories (if this expansion cannot be restricted or discouraged). For instance, our results show that the average number of tsunami-caused deaths would occur across those evacuees initially located within an approximately 300-meters-wide buffer zone from the coastline. In line with this, Løvholt et al. (2014, p.133) point out that studies of the impact of the 2004 Indian Ocean Tsunami show that "in Sri Lanka, people within the 100-m zone from the shoreline were more likely to die and to be seriously injured than people living outside this zone". In turn, González-Riancho et al. (2015) underline that 72% of the housing units within the 200-m line from the shoreline in Sri Lanka were completely or partially damaged, leading to a higher number of victims. Eckert et al. (2012) also point out that buildings within that area are highly vulnerable. In the case of the tsunami flood, while inundation depths can be above 10 meters at several of our case studies' coastlines, our model shows that the average flood depth at the 'lethal' departure cells is roughly 4.67 m. In turn, 'safe' cells have a comparatively low mean value of 0.61 m, implying that some evacuees can avoid being caught by the advancing tsunami front if they rapidly leave the low floodable areas. These results are in line with the literature on human casualties during past tsunamis. For instance, Suppasri et al. (2016) point out that the inundation depths that increased fatality ratios during the 2011 Great East Japan Tsunami are primarily around 10 or 5 m, depending on the specific geographical characteristics of different examined areas. In line with this, Murakami et al. (2012)

examined the human loss distribution during the 2011 disaster in Yuriage District, Natori City, showing that inundation depths between 1.87 m and 8.50 m triggered death ratios up to 22.3%. In turn, in the case of the ground elevation, the average value of the 'safe' cells is 13.82 m. When we include 'lethal' cells in the analysis, we can see that fatalities concentrate around elevations of six meters and below. In this respect, Eckert et al. (2012) argue that buildings located at a height of 5-10 m can be considered of medium vulnerability to tsunamis, while those with an elevation above 10 m have low vulnerability. For the previously mentioned case of the Yuriage District, Murakami et al. (2012) report elevations less than five m in the deadly areas. It is also noticeable that cells belonging to street networks with good integration levels, that also allow short walks to the safe assembly areas, and have few direction changes (i.e. with high betweenness, straightness and PDR values, respectively), have lower death ratios. In this respect, as Sharifi (2019) underlines, locations with high betweenness centrality values can easily lead to many other sites within the network. Therefore, it is critical to maintaining their functionality during disasters.

To focus on possible paths to improvement for these case studies it is helpful to examine the outcomes from our multivariate regressive analysis. In this respect, as we pointed out above, some independent variables have comparatively higher impacts on the death ratios as predicted by our regressive model. The most significant one is the maximum flood depth, followed by the straightness, the total route length, and the travel time. The maximum floods on 'lethal' cells are difficult to mitigate unless hardware-type defences are built (which, as mentioned above, is unlikely in developing countries like Chile). Moreover, we already pointed out that the tsunami flood also affects the travel time in our regressive analysis. Nevertheless, the straightness and total route length depend on real-world urban configurations, resulting from the case studies' historical development process. They hence can be subject to strategic interventions to modify their values to reduce the cells' death ratios. In this respect, more direct routes are not only faster to walk (thus reducing escape distances and evacuation times) but also help to improve wayfinding as they reduce the changes of directions that evacuees must undertake between their origins and destinations (Fakhrurrazi and Van Nes, 2012; Mohareb, 2011). The importance of wayfinding cannot be underestimated, especially in the case of tourists and non-locals, who may constitute a large percentage of casualties during a tsunami (as shown by the Chilean disaster of 2010) (Kubisch et al., 2020). Also, the total route length (and the shelter distance) could also be reduced by incorporating vertical evacuation across the urban fabric, which has been proven to reduce the evacuation times significantly (León et al., 2019; Mostafizi et al., 2019). Currently, vertical evacuation in Chile is recommended as only a second choice of escape if horizontal evacuation is not feasible (ONEMI, 2014).

Thorough evacuation analyses are context-dependent and must take care of geographical and socio-psychological aspects that affect the populations' behaviour (Makinoshima et al., 2021; Mohareb, 2011; Murray-Tuite and Wolshon, 2013; Perry et al., 1981). In this respect, one limitation of our study is that socio-psychological factors are not analysed. However, future research on this type of determinants of tsunami evacuation could help to critically review some of our model's central assumptions (e.g. a 'full compliance' evacuation, the probabilistically distributed departure times, or the routing process) and strengthen future outputs. Furthermore, as Suppasri et al. (2016, p.11) point out, "analyses involving statistically significant correlations between characteristics and fatality rate must be performed with caution and based on various data sources". We

are aware that our findings' reliability depends on the quality of the model's assumptions, functions, and source data. In this respect, reality-based validation procedures (as the one mentioned in León et al. (2021a)) will always be necessary, and the related spatial planning guidelines for evacuation improvement should be delivered cautiously. Nevertheless, as tsunamis are relative rare phenomena (where populations' actual behaviours are still hard to capture), our simulation-based analysis provides a significant step into identifying and examining geographical and built environment's attributes that might influence the evacuation potential of coastal communities, as a spatial framework for the subsequent analysis of their specific socio-psychological characteristics.

## 5 Conclusion

- We proposed a modelling-based approach (including inundation, evacuation, and urban form metrics) to quantitatively appraise, through statistical regressive analysis, some of the most relevant aspects of the geographical and built environments that could contribute to the success (or failure) of evacuation in the case of a tsunami, using a cross-case study of seven Chilean coastal cities.

- According to our results, some of these cities can have up to roughly 55% of their move boundaries (i.e. the evacuation area between the coastline and the safe inland assembly areas) susceptible to having dead evacuees.

- We also demonstrated that geographical, urban form and evacuation variables, including the maximum flood depth (within the examined evacuation threshold), straightness, total route length, and mean travel time, could significantly impact the expected death ratios in each case study. Moreover, we describe the average values of these metrics related to different thresholds of death ratio.

- We argued that, while engineered countermeasures to control flood levels are unlikely in developing countries like Chile, urban form metrics like the street network's straightness could be the subject of improvements through planning processes. Moreover, this would allow other enhancements in other evacuation dimensions like the travel time and evacuees' wayfinding.

- Future research could enhance our approach with the incorporation of socio-psychological aspects and probabilistic tsunami flood modelling. Also, more case studies (at both the national and global levels) and validation procedures could help test our findings' robustness and generalizability.

## Data availability

DBF files compressing the post-processed spatial data can be downloaded from: https://usmcl-my.sharepoint.com/:f:/g/personal/jorge_leon_usm_cl/Emfj8ZYFmb1EhZmQ49dbLJ4Bi5z1HOvc-TKHsg0Wg5GZBw?e=KAL2Z5

**Author contribution**

Jorge León: conceptualisation, methodology and writing – original draft. Alejandra Gubler: tsunami and evacuation modelling, review and editing. Alonso Ogueda: statistical modelling.

**Competing interests**

The authors declare that they have no conflict of interest.

**Acknowledgments**

This research was funded by the research grants ANID/FONDECYT n∘ 11170024 and 1210184, and by the National Research Center for Integrated Natural Disaster Management (CIGIDEN), ANID/FONDAP/ 15110017. Powered@NLHPC: This research was partially supported by the supercomputing infrastructure of the NLHPC (ECM-02). We also thank Matías
Carvajal for providing the fault model of the Iquique earthquake.

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
