# Peer review of "Modelling geographical and built environment's attributes as predictors of human vulnerability during tsunami evacuations: a multi-case study and paths to improvement"

_Natural Hazards and Earth System Sciences, 2021_

## Author Response (AR1)

**Response to the reviewers' comments to authors**

We thank the editor in charge for the opportunity to submit a revision of our manuscript. We also thank the reviewers for providing feedback for improvement. We have responded in detail to each comment in the following pages, and we think that these amendments have significantly strengthened our paper.

We believe that the new version of our manuscript is a significantly improved one and more readable for broader audiences.

**Reviewer # 1**

We thank the reviewer for his/her comments. It follows detailed replies to each of them.

- The authors used various variables to explain the simulated degree of tsunami casualties. Because some of the explanatory variables seems to have correlations among them, I have had concerns the technical problem of statistical methods such as collinearity/multicollinearity problems. Since inappropriately constructed statistical models can lead to wrong results, I suggest additional validity checks of the statistical method. Additionally, the justifications including detailed explanations of the methods and the choice of the methods from similar regression models can help readers' better understanding.

  **A:** We included a correlation test to prevent the correlation between the death ratio (our dependent variable) and the other likely predictor (independent) variables, to avoid collinearity problems in the regressive model. We use the results of this test to select only nine variables for further analysis. These changes were included in section 2.2.4, starting in page 11, line 266 (old manuscript). We also included further explanation about the random forest methodology and the SHAP (SHapley Additive exPlanations) model, starting in section 2.3, page 11, line 277 (old manuscript).

- Different from the previous statistical models explaining tsunami casualties based on actual data, the statistical models were constructed only based on the simulation results from agent-based simulations. Although the method has an advantage that various data can be generated from simulations, at the same time, the quality of the data and the constructed statistical is totally depends on the quality of the simulation. Therefore, inappropriate modellings or excessive speculation from the results can lead to inappropriate implications for actual evacuation preparedness.
  Along with the confirmation of the validity of the method and data itself, discussions in this paper should supported by additional validity check using simulations, and the applicability of the results should be carefully discussed.

  **A:** We thank the reviewer for his/her comment. We included a reference to a recently published paper where we validate our agent-based model using real-world data (section 2.2.1, starting in page 8, line 181 of the old manuscript). We agree that issues about the reliability and generalization of the results should be carefully discussed. With this purpose, we enhanced the Discussion section, starting on page 15, line 385 (old manuscript).

- p.1 Line 26 - 29: The discussed integrated approach for tsunami disaster risk reduction is summarised well with its historical transitions in Koshimura & Shuto (2015), which would be useful to support the description. The paper can be found at https://doi.org/10.1098/rsta.2014.0373

  **A:** We used this new reference to enhance the manuscript in page 1, line 27 (old manuscript).

- p.2 Line 31 - 33: The authors claim "this is hard to achieve...", but the reason why is not well expressed. It is better to make it clear for readers from broader research fields.

  **A:** We included further explanation about this topic in page 2, line 31 (old manuscript).

- p.2 Line 35: What information brings the value "15 min"? In my opinion, the evacuation behaviour is always an effective way to save lives during tsunamis if there is sufficient lead time.

  **A:** We agree with the reviewer. However, as our primary focus is on the Chilean case, we underline that the country's typical short arrival times put strong pressure on evacuation processes. We included further explanation and references about this topic in page 2, line 35 (old manuscript).

- p.2 Line 37 - 39: In my understanding, "hazard" simply represents an intensity of external force and is not the term to represent how existing condition is affected. For example, this page (https://www.preventionweb.net/understanding-disaster-risk/component-risk/disaster-risk) explains hazard as "Hazard is defined as the probability of experiencing a certain intensity of hazard (eg. Earthquake, cyclone etc) at a specific location and is usually determined by an historical or user-defined scenario, probabilistic hazard assessment, or other method. Some hazard modules can include secondary perils (such as soil liquefaction or fires caused by earthquakes, or storm surge associated with a cyclone).", with the source GFDRR, 2014. Such terminology should be consistently used, referencing reliable in official documents.

  **A:** As suggested by the reviewer, we included further references from official sources to strengthen our concepts definitions. Page 2, from line 36 (old document).

- p.2 Line 47 - 50: In my view, some items are inappropriately categorized. For example, is "elevation" exposure? Again, the abovementioned page defined the exposure as "Exposure represents the stock of property and infrastructure exposed to a hazard, and it can include socioeconomic factors". It is better to categorise them with an exact criterion, referencing corresponding sources.

  **A:** We included the new word "determinants" to underline that in this sentence we are referring to those factors that contribute to either increase or decrease the exposure characteristics. The paragraph was also enhanced with further words. Page 2, line 47 (old manuscript).

- p.2 Line 55: "Fragility function" is often used in this context.

**A:** We agree with the reviewer. We modified the text accordingly. Page 2, line 55 (old document).

- p.3 Line 67 - 90: This part lines up the existing literature regarding fragility functions for tsunami casualties. Is there any criterion regarding the order of these literatures? It started from the study in 2018 and goes to 2020, but it then suddenly back to 2009. Since these studies develops their method, usually referencing old ones, it is better to present them as readers can understand the trend of these studies. If there is an intention of authors for this order, the text flow should be modified to make it clear. Additionally, the review seems to lack some literature in the same line. Additional reviews would be useful to be more comprehensive. For example, Suppasri et al., 2016 (https://doi.org/10.3389/fbuil.2016.00032); Latcharote et al., 2018 (https://doi.org/10.1016/j.ijdrr.2017.06.024); Yun et al., 2019 (https://doi.org/10.1193/082013EQS234M).

  **A:** We agree with the reviewer. We entirely modified this section of the manuscript, also including additional references. From page 3, line 67 (old document).

- p.3 Line 93 - 94: The references seem to lack recent literature, especially which simulates detailed 2D evacuation movements. Additional literature review would help to make it more comprehensive. For example, Dohi et al., 2016 (https://doi.org/10.1142/S1793431116400108); Aguilar and Wijerathne, 2016 (https://doi.org/10.1142/S1793431116400212); Makinoshima et al., 2018 (https://doi.org/10.1016/j.simpat.2017.12.016).

  **A:** We modified the text to enhance the manuscript, including the suggested additional references. From page 3, line 93 (old document).

- p.3 Line 95: There are references that are better to be refereed here for the review of GIS-based methods. For example, Fraser et al., 2014 (https://doi.org/10.5194/nhess-14-2975-2014); Priest et al., 2016 (https://link.springer.com/article/10.1007%2Fs11069-015-2011-4).

  **A:** We modified the text to enhance it, including the suggested additional references. From page 3, line 95 (old document).

- p.4 Line 102 - 110: Since the there are tremendous amount of literature regarding tsunami evacuations, the reference here seems insufficient. Recent comprehensive review of tsunami evacuation behaviours would be useful for supporting the discussion here. The review paper, Makinoshima et al., 2020, can be found at https://doi.org/10.1016/j.pdisas.2020.100113

  **A:** We entirely modified this section of the manuscript, also including additional references. From page 4, line 102 (old document).

- p.5 Figure 1: This can be moved to the next Methodology section because detailed explanation was made in the next section, and only the names of cities are described in the first section.

  **A:** We agree with the reviewer. We moved Fig. 1 to the Methodology section.

- p.6 Line 143: Does "35 destructive events" include "recent disasters in 2010, 2014 and 2015" ? The text can be modified for clarity.

  **A:** We modified the text to make clear that the 35 destructive events count to 2005. Page 6, line 143 (old document).

- p.6 Line 146 - 147: Here is a suitable place to present the figure 1.

  **A:** We thank the reviewer for his/her suggestion. Indeed, we moved Fig. 1 to this suggested place.

- p.6 Line 154: I would say "repeatedly" instead of "systematically".

  **A:** We modified the text to include this change. Page 6, line 154 (old manuscript).

- p.7 Table1: It is better to present the items in "Years of recorded destructive tsunamis" with its event name and references for its mechanisms since easily accessible information of the events would be useful for readers. The table captions should be presented at the top.

  **A:** We modified Table 1 to include references to the included earthquakes. The table caption was moved to the top, too. Page 7 (old manuscript).

- p.7 Line 161: I understand that this resolution "4x4 m" is based on the finest resolution of the tsunami simulation; however, this resolution might too fine for counting tsunami casualties in agent-based simulations. The investigation with different resolution is needed to ensure the validity of the result. If consistent important features are found in different resolution, it supports the validity of the analysis method. Reliable coarser values can be generated by integrating finer values.

  **A:** We understand the suggestion made by the reviewer. However, our modelling technique used the STOC software, which couples tsunami and evacuation models, therefore using the same grid with a unique resolution. This feature makes unfeasible to execute the agent-based model independently using a different grid size. Nevertheless, as we pointed out above, the evacuation model was validated with real-world data in a recently published paper.

- p.8 Line 163 - 175: The figure showing the simulation setup for tsunami simulations (e.g., visualization of fault shape and its slip amount in a map) would be useful for better understanding of readers.

**A:** We included a new figure (Figure 2 in the new manuscript) showing the seismic simulation setups for tsunami modelling. Page 8, line 175 (old document).

- p.8 Line 178: Which part of the simulation was "enhanced" compared to the original model? It should be clear.

  **A:** We included further explanation about how the source code was enhanced by us.

- p.8 Line 181 - 182: This description is true for agent-based modelling that simulates detailed interactions among agents (e.g., social force model), and I think readers expect this study used such model after reading this description; however, the detailed explanation of the model (p.9 Line 203 - 215) explains that the model does not simulate such complex interactions (e.g., speed down due to the congestions, which caused by detailed 2D behaviour simulations). The text should be modified to more clearly express the model capability.

  **A:** We modified the text according to the suggestion by the reviewer. Now, we hope to clearly express the model capability, focused on a macroscopic perspective.

- p.9 Line 183: Previously mentioned recent evacuation models, which simulates 2D detailed movements and complex interactions (Dohi et al., 2016 (https://doi.org/10.1142/S1793431116400108); Aguilar and Wijerathne, 2016 (https://doi.org/10.1142/S1793431116400212); Makinoshima et al., 2018 (https://doi.org/10.1016/j.simpat.2017.12.016)), is better to be refereed in this context.

  **A:** We modified page 3, from line 92 (old manuscript) to include the suggested references.

- p.9 Line 204 - 205: It is unclear whether "a mean time = 8 min" is the mean value of the resulting distribution or a parameter value for the probability distribution. If "8min" refers to the parameter sigma for Rayleigh distribution, the resulting mean value of the distribution does not match this value. Presenting the mathematical expression of the Rayleigh function with the parameter used in this study can avoid any confusion.

  **A:** We included a new reference (Mas et al., 2012) and further text to avoid confusion. Page 9, line 204 (old manuscript).

- p.9 Line 211: In my understanding, the paper cited here is not the paper that first proposed the A* algorithm. Is there any reason to cite this paper here? For example, if the study used the implementation in the citing literature, the authors should explain so to avoid any ambiguity.

  **A:** We agree with the reviewer. We edited the text and included further references focused on evacuation studies that apply the algorithm. Page 9, line 211 (old manuscript).

- p.10 Line 217 - 218: The number of required simulation runs should be reported here instead of explaining "at least ten" because the information is useful for readers to know the simulation variance.

  **A:** We ran 10 simulations for each case study. We edited the text to avoid ambiguity. Page 10, line 217 (old manuscript).

- p.7 Line 158 - p.10 269: Sections 2.x.x includes both methods to generate data and metrics, and this mixed description can lower the readability of the manuscript. The structure of these sections can be re-structured for better readability.

  **A:** We appreciate the reviewer's comment. However, we prefer to keep the structure of these sections in its current shape.

- p.11 Line 251 - 269: As I pointed out for p.7 Line 161, it is better to conduct the analysis with different resolution to see the effect of resolution and check the validity of the analysis.

  **A:** Please see above our reply to this subject.

- p.11 Line 275 - 277: Although the random forest is a popular regression model, it is better to explain what it actually is with its brief theoretical explanation for completeness of this paper and better understanding of readers.

  **A:** We included further references and new text to expand the explanation of the random forest methodology. Page 11, line 277 (old document).

- p.12 Line 287: I think the brief theoretical explanations on SHAP and SHAP values is necessary because most of the readers in this field are not familiar with this. At least, readers have to know the logics to estimate the importance of the explanatory variables in understanding the results presented in the rest of this paper.

  **A:** We included further references and new text to expand the explanation of the SHAP values methodology. Page 12, line 289 (old document).

- p.13 Table 3 and Figure 2: The table and the figure simply display the mean value of the variables and is not accompanied with any detailed explanations and discussions such as regional differences. As a result, current text and materials carry almost no information to readers. The reported value can be improved by reporting variances so that readers know the distribution of the variables. Instead of tables and the current figure, a scatter plot matrix representation of the raw data may be useful for understanding data. Along with the revised visual representation, detailed description of the general tendency of the data is required in the revised manuscript.

  **A:** We modified Table 3 to include, for every examined variable and death ratio threshold, the standard deviation. Also, we included a new Figure 5, showing data scatterplots with the distribution of the death ratios, in comparison to the nine examined

independent variables. We also deleted the old Fig. 2. From page 12, line 301 (old document).

- p.14 Line 307 - 312: Because some explanatory variables potentially have correlations (e.g., Maximum flood depth and Elevation; Straightness, Route length and Mean travel time), I wonder if the analysis has the problem such as collinearity/multicollinearity. The previously mentioned scatter plots of raw data can help readers to consider such potential problems in data. Although such correlations may not affect the performance of the constructed model, I think it at least affect the value of importance. Confirmation of the data and the justification of the validity of the result are required. Additionally, this section simply present figures and no in-depth explanations are made. Broader implications of the results or relations to the previous literature can be presented in the following Discussion section; however, this section at least should describe the obtained results in detail.

  **A:** As we pointed out above, we included a test to prevent the correlation between the death ratio (our dependent variable) and the other likely predictor (independent) variables, to avoid collinearity problems in the regressive model. Overall, we enhanced the Results section to provide an in-depth explanation of our findings. From page 12, line 299 (old document).

- p.14 Line 326 - p.15 Line 341: This part discusses positive and negative factors influencing the simulated level of tsunami casualties. Although the all explanatory variables are not always meaningful to explain the target variables, the discussions are presented only based on a qualitative view. Because the authors conducted statistical analysis and know the importance of explanatory variables, such discussion should be made considering whether the variables are statistically meaningful.

  **A:** We enhanced the Discussion section to include a more quantitative approach to the reviews of our findings. From page 14, line 327 (old manuscript).

- p.15 Line 332 - 341: This part explains counterintuitive results and its potential cause; however, in my view, these explanations need further validations because the data is synthesised using simulations, and the data might be generated from unintended behaviour of the simulation models. For example, combination of very local error in elevation data and the hiking function may cause unrealistically slow evacuees. Because the observed tendency is generated from data in simulations, the authors can validate their explanations by checking the simulation results in detail. Cause of the synthesised data can be clearly explained by simulations, and should be.

  **A:** We agree with the reviewer. We enhanced the Discussion section to address his/her comments. We included a new reference to our recently published paper, which validates our evacuation model with real-world data. We also add new text that discusses the limitations of our results. From page 14, line 317 (old manuscript).

- p.15 Line 338 - 339: For example, this description should be supported by showing such simulation results.

**A:** We agree with the reviewer. However, rather than being on each case study's specific characteristics, the paper's focus is on the 530,091 examined cells.

- p.15 Line 343 - 345, Line 345 - 346, Line 353 - 354: Since the simulations in this study does not include realistic evacuation processes and are based on various assumptions (e.g., a single evacuation departure distribution), it is hard to reach general conclusion using this approach. Such limitations should be clearly expressed, and any extrapolation of the results may lead to proposing inappropriate guidelines.

  **A:** We included further discussion about these limitations. From page 15, line 385 (old document)

- p.16 Line 386 - 387: Recent study reported that tsunami evacuation processes are largely affected by socio-psychological factors and exhibit complex evacuation trips. Referencing such example would support the claim. For example, Makinoshima et al., 2021 (https://doi.org/10.1016/j.ijdrr.2021.102182).

  **A:** We modified the text and included the suggested reference. Page 16, line 386 (old manuscript).

**Reviewer # 2**

We thank the reviewer for his/her comments. It follows detailed replies to each of them.

- In general, the paper is well written, and methodology and results are clearly presented. However, introduction of 4 pages seems to be too large and makes it difficult to identify the main focus and scientific gap to be researched. It would be convenient to shorten the introduction. In addition, since the main topic is the human vulnerability, I suggest to shorten the paragraphs from lines 36 to 66, since specific explanation of fragility curves may not be necessary.

  **A:** We understand the reviewer's suggestions. While we decided not to shorten the Introduction section (as we considered relevant to deliver a comprehensive review of the state-of-the-art on tsunami human vulnerability), we enhanced this section with further references, and a more specific focus on how this vulnerability relates to the attributes of the geographical and built environments.

- Figure 1 should be in methodology. A new figure 1 should show a map of South America, Chile and the topo-bathymetry of each location in order to have an idea of the morphology of each city.

  **A:** We moved Fig.1 to the Methodology section. We enhanced this figure to include the location of Chile in South America, the topo-bathymetry of each examined city, and the estimated tsunami arrival times. Page 6, line 147 (in the old manuscript).

- The Figure 1 (new figure 2 in section 2.1) should show the tsunami arrival time instead of just the inundation area.

**A:** We modified Figure 1 to also include (for each case study) the estimated tsunami arrival times. Page 6, line 147 (in the old manuscript).

- Section 2.2.1 should include a figure with example of simulation grids and all tsunami scenarios used in the analysis.

  **A:** We included a new Figure 2 to show the simulated seismic scenarios used for the case studies. In page 8, line 175 (old manuscript).

- Even though the resolution of the tsunami simulation is 4m, it would not be necessary to use the same resolution for the agent-based simulations. Since the inundation was recorded every 10 min (line 174), this measure can give you a necessary resolution for agent-based simulations. In fact, several resolutions may be used and similar results should be obtained.

  **A:** We understand the suggestion made by the reviewer. However, the modelling technique used the STOC software, which couples tsunami and evacuation models, therefore using the same grid with a unique resolution. This feature makes unfeasible to execute the agent-based model independently using a different grid size.

- Line 174 indicates that numerical model record the time series, however this results are not shown in the paper. The tsunami wave forms are also important to analyze the tsunami arrival time and whether you captured the maximum inundation. Please add a figure to show those time series.

  **A:** We included a new Figure 3 showing the tsunami time series, for each case study. Page 8, line 177 (old manuscript).

- Line 172. Please clarify why only 45 min of elapsed time was used. It is well known that Talcahuano has some resonant effect and maximum tsunami inundation take place after several hours. In addition, it has been observed that the second or third wave are usually the largest one.

  **A:** We understand the suggestion made by the reviewer. Indeed, local effects may lead to several hours of sea level anomalies. Nevertheless, preliminary modelling tests showed that evacuations in every case study would not take more than 36 minutes to complete. Therefore, we set up a 45-minute threshold to conservatively encompass the total evacuation process, regardless of longer tsunami effects. We included further explanation of this in the manuscript. Page 8, line 173 (old manuscript).

- Line 322. Only 0.74% of cells in Talcahuano show to have elevated dead ratio. It is not unexpected since the inundation given in figure 1 is not that large. What would be the result if you analyze the maximum tsunami inundation instead of just 45 min?

  **A:** As we pointed out above, our methodology uses a coupled tsunami-evacuation model where both phenomena develop along the same timeline. Therefore, if the

maximum tsunami occurs after the evacuation was complete (which in the case of Talcahuano takes roughly 27 minutes after the tsunamigenic earthquake), no extra casualties will occur. Of course, this might change if we change the model's assumptions, such as the evacuation departure rate.

- Line 340. I understand that from an evacuation point of view you are interested in the first tsunami front. This may explain why you used only 45 min. However, in 45 min, some areas may have 2 or 3 tsunami waves, while Talcahuano would have only one. It would be necessary to use the same criterion for all locations.

  **A:** Please see our answers above.

- Lines 346. It is observed that average number of casualties would occur within 300 m from coastline. It would be interesting to analyse the effect of distant to trench, due to the fact that this variable affect tsunami propagation and subsequently the tsunami arrival time. Therefore, cities in northern Chile would experience larger number of tsunami-caused deaths than cities in central or southern Chile. Is that correct?

  **A:** We understand the suggestion made by the reviewer. It is possible that other geomorphological variables, like the distance to the trench, can influence the expected death ratios. Nevertheless, it is hard (without a full new statistical analysis) to isolate the importance of that single variable, having into account that (as our analysis showed) other features (like the street network configuration) also vary from case to case and have an impact on evacuation times and, therefore, the survival rate. We included in the Discussion section a list of other variables that could be included in further analyses. Page 16, from line 385 (old manuscript).

- Line 399 indicates that maximum flood was analysed, however, with 45 min of simulation some cities may not reach the maximum inundation. As indicated in comment 5, Please add a figure of tsunami waveforms in the results section in order to show that maximum tsunami flood was analyzed.

  **A:** We included the requested new Figure 3 (page 8, line 177, old manuscript). Also, we included changes throughout the manuscript to underline that we examine the maximum flood within the evacuation threshold of 45 minutes.

---

## Author Response (AR2)

**Response to the reviewers' comments to authors (second revision)**

We thank the editor in charge for the opportunity to submit a new revision of our manuscript. We also thank the reviewers for providing feedback for improvement. We have responded in detail to each comment in the following pages, and we think that these amendments have significantly strengthened our paper.

We believe that the new version of our manuscript is a significantly improved one and more readable for broader audiences.

**Reviewer # 1**

We thank the reviewer for his/her comments. It follows detailed replies to each of them.

- Most of my concerns have been addressed in the revised manuscript. In the previous round, I suggested to check the validity of the results by conducting statistical analysis with different resolutions; however, the revised manuscript had not addressed this comment. For this response, authors explained that they could not conduct additional analysis due to their software specification, though, in my understanding, coarser values for additional analyses can be generated from current finer results. Since the comment concerns the effect of resolution to collect statistics from the simulations, in my view, the response regarding the validity of their simulation model cannot help to address this concern. Additional analysis with different resolutions is better to be included to ensure the validity of the obtained results. At least, discussion regarding the potential effect of the model resolutions is better to be presented.

During the first review, the reviewer pointed out that

- I understand that this resolution "4x4 m" is based on the finest resolution of the tsunami simulation; however, this resolution might too fine for counting tsunami casualties in agent-based simulations. The investigation with different resolution is needed to ensure the validity of the result. If consistent important features are found in different resolution, it supports the validity of the analysys method. Reliable coarser values can be generated by integrating finer values.

  **A:** To answer the reviewer's comment, we included a new SHAP values analysis with the same source information but combined in different resolutions, according to a geometric sequence with a factor of 2 between the size of each examined unit. To do this, we developed an algorithm that, starting from the 4x4 m cell (resolution x1) with the largest southern latitude and western longitude in each case study (i.e. located at the bottom left corner of the study area), grouped these basic units into five successive spatial partitions, each of them covering the complete evacuation move boundary. These partitions comprised cells 8x8 (resolution x2), 16x16 (resolution x4), 32x32 (resolution x8), 64x64 (resolution x16), and 128x128 m (resolution x32) wide, respectively. For each of these larger cells, the algorithm calculated the value of every independent variable as the average of the combined 4x4 m units. Then, we ran again the SHAP values analysis at these coarser resolutions. The results (summarized in the new Figure 8) show that the overall importance hierarchy of the independent variables remains unchanged through the different resolutions of analysis. Complementarily,

noticeable and disparate changes can be seen in the amount of their impacts on the predicted death ratio, if we compare the more and less accurate resolutions. We also discuss these methods and findings in the "Discussion" section. Page 18, line 371, page 23, line 441, page 26, line 530, and page 27, line 563.

**Reviewer # 2**

We thank the reviewer for his/her comments. It follows detailed replies to each of them.

- In line 232, it says that "Tsunamis were simulated for 45 min...". However, figure 3 shows that tsunamis in most of the study sites were simulated for 60 min, except Valparaiso-Vina del Mar. it should be coherent.

  **A:** This contradiction appeared in the manuscript because we first ran the flood models for 60 minutes (except in Valparaíso-Viña del Mar) and afterwards we used them as inputs for the agent-based simulations, which we ran for only 45 minutes, as this was sufficient to encompass the total evacuation time of each case study (which also allowed us to save computing time). We made this point clear in the new version of the manuscript. Page 10, line 204, and page 13, line 218.

- Figure 2. The scale of the vertical displacement is confusing. In one scenario, the vertical displacement of 7m is high, but in the other 4 m is high. Instead of using a colour scale with high or low, just use numbers. In addition, it could be convenient to use the same scale for all scenarios, thus they can be compared.

  **A:** We re-worked the figure accordingly. The new figure 2 (page 11) uses the same colour scale for all scenarios. Additionally, this scale includes numerical intervals.

---

## Author Response (AR3)

**Response to the reviewers' comments to authors (final acceptance)**

We thank the editor in charge for accepting the new revised version of our manuscript for final publication in NHESS. We also thank the reviewers for providing their critical feedback for improvement. We think that responding in detail to each of their comments allowed us to strengthen our paper significantly.

Warm regards,

**Dr Jorge León**